# Management of patients with an advance decision and suicidal behaviour: a systematic review

Rebecca Nowland,[1] Sarah Steeg,[1] Leah M Quinlivan,[1,2] Jayne Cooper,[1] Richard Huxtable,[3] Keith Hawton,[4] David Gunnell,[5] Neil Allen,[6] Kevin Mackway-Jones,[7] Navneet Kapur[1,2,8]

For numbered affiliations see end of article.

**Correspondence to**
Dr Rebecca Nowland;
rnowland@uclan.ac.uk

## ABSTRACT

**Background** The use of advance care planning and advance decisions for psychiatric care is growing. However, there is limited guidance on clinical management when a patient presents with suicidal behaviour and an advance decision and no systematic reviews of the extant literature.

**Objectives** To synthesise existing literature on the management of advance decisions and suicidal behaviour.

**Design** A systematic search of seven bibliographic databases was conducted to identify studies relating to advance decisions and suicidal behaviour. Studies on terminal illness or end-of-life care were excluded to focus on the use of advance decisions in the context of suicidal behaviour. A textual synthesis of data was conducted, and themes were identified by using an adapted thematic framework analysis approach.

**Results** Overall 634 articles were identified, of which 35 were retained for full text screening. Fifteen relevant articles were identified following screening. Those articles pertained to actual clinical cases or fictional scenarios. Clinical practice and rationale for management decisions varied. Five themes were identified: (1) tension between patient autonomy and protecting a vulnerable person, (2) appropriateness of advance decisions for suicidal behaviour, (3) uncertainty about the application of legislation, (4) the length of time needed to consider all the evidence versus rapid decision-making for treatment and (5) importance of seeking support and sharing decision-making.

**Conclusions** Advance decisions present particular challenges for clinicians when associated with suicidal behaviour. Recommendations for practice and supervision for clinicians may help to reduce the variation in clinical practice.

## INTRODUCTION

An advance decision (sometimes known as an advance decision to refuse treatment or living will) is typically a written document that outlines a person's desire to refuse certain treatments, including life-saving treatment, when there is a potential for a person to lose the mental capacity to make treatment decisions in the future.[1] In order for an advance

decision to be valid, the person must have mental capacity at the time of writing the document. Mental capacity is defined as the ability to make a decision and involves understanding and weighing information relating to a decision and alternative options and retaining that information long enough to make the decision.[1] The Mental Capacity Act in England and Wales refers to 'advance decisions to refuse treatment', but more widely these documents are referred to as 'advance directives' and/or 'living wills'. We use 'advance decision' throughout in this paper to refer to written documents stating a refusal of treatment made in advance of medical treatment following an illness or injury.

There are important cross-national variations in legislation; in some countries, the use of advance decisions is not permitted (ie, Turkey, Japan), while in others, advance decisions are legislated for (ie, the UK and USA). The UK, Australia and USA have similar legal standards with some state-wide variation in the USA and Australia,[2] with some states adopting the common law right to make an advance decision and others allowing the use of a surrogate or proxy decision-maker (ie, to make healthcare decisions on behalf of the patient). There is also considerable variation in practice between countries where advance decisions are permitted. For example, in

**BMJ**

Germany, advance decisions are recognised but require court approval in each case.[2]

Advance care planning for psychiatric care is becoming more common in a number of countries, including the UK, USA and Australia[3 4] and enables patients to state their preferences for the management of their mental health when they may temporarily lose their mental capacity. A person with a mental health disorder may also make some decisions about particular treatment that they would not wish to have and may involve an advance decision to refuse particular treatments (ie, electroconvulsive therapy). Advance care planning has been shown to have a number of healthcare benefits for mental health patients in the UK and USA, such as enhancing patient autonomy and engagement, promoting adherence to treatment plans (ie, patients taking prescribed drugs), improving continuity of care with fewer psychiatric admissions, reducing the use of social workers' time and lower levels of violent acts.[3 4] In a recent survey of patients with bipolar disorder, 21% had written statements about their healthcare, and of those, 10% involved an advance decision.[5] This increasing use of advance care planning in mental health may result in an increasing use of advance decisions to refuse mental healthcare treatment, and concerns about clinical management of advance decisions following suicidal behaviour have been made by healthcare professionals and legal and ethical consultants.[6–8] Existing literature, from a variety of academic and clinical perspectives, suggests there is little consistency in practice, and there are specific challenges with advance decisions following suicidal behaviour. Such scenarios raise questions about whether a person with a wish to end their life has the capacity to make a decision about refusal of treatment and/or if their capacity is affected by mental illness, and whether an advance decision is appropriate for medical treatment following suicidal behaviour.[8]

The terminology for suicidal behaviour varies internationally. Some clinicians/researchers distinguish between suicide attempts and non-suicidal self-injury,[9] while others prefer the broad term of self-harm to denote behaviours across the spectrum.[1 10] We have taken an inclusive approach in this review to ensure we captured relevant studies, so in this review we refer to 'suicidal behaviour' as behaviours including all self-harming behaviour (including non-suicidal injury) and suicide attempts. The use of 'suicidal behaviour' in our review means that there may be cases of non-suicidal injury that were included.

The management of suicidal behaviour is a significant challenge for clinicians in the emergency services. Each year over 200 000 people present to emergency departments (EDs) in England with self-harm,[10] with 16% of those presenting to hospital with a repeat self-harm episode within a year.[11] Treatment refusal following suicidal behaviour has been shown to be common. A prospective cohort study of mental capacity and suicidal behaviour in the ED found that around 40% of patients presenting to hospital with self-harm had the capacity to make a decision about their medical treatment and 30%

of those intended to refuse life-saving treatment.[12] There are few studies that have examined numbers of advance decisions to refuse treatment in patients presenting with suicidal behaviour, but in a recent study in three of 121 fatal cases of self-poisoning in 2005, patients had an advance decision.[13] Given that patient autonomy and advance care planning are encouraged in modern healthcare and are assuming greater prominence, it is likely that the number of people presenting to hospital with an advance decision following suicidal behaviour will grow.

### Rationale

While reviews of literature relating to the management of advance decisions, both more broadly and specifically to relating to 'end-of-life' care exist,[14 15] there are currently no reviews on the management of advance decisions when a patient presents to hospital following suicidal behaviour where the patient does not have a chronic or terminal physical illness. Despite the legislative context being similar for end-of-life care, the ethical considerations, emotional challenges and clinical decision-making may be different for treatment of a patient following suicidal behaviour without a chronic or terminal physical illness. A synthesis of this literature is important to examine similarities and differences and to establish the key findings. This is particularly important as the management of advance decisions to refuse treatment of injuries and illnesses following suicidal behaviour is challenging for clinicians[8] and there is a lack of consistency of practice. A review of the literature will be important to inform guidelines for the management of advance decisions following suicidal behaviour.

### Aim

To systematically review and synthesise literature on the treatment and clinical management of patients presenting to hospital with an advance decision to refuse treatment following suicidal behaviour without a chronic or terminal physical illness. The review was conducted by researchers in the UK, but an examination of all the existing literature was conducted without language or country restrictions.

## METHOD

The review was conducted in accordance with Preferred Reporting Items for Systematic Reviews and Meta-Analyses (PRISMA) guidelines[16 17] and guidance for conducting narrative synthesis in healthcare.[18] There is no protocol for the review. We used the PRISMA checklist when writing our report.[16]

### Search strategy and data sources

An initial scoping of the literature was conducted at inception of the study and the findings were used to inform the search strategy. Content experts and clinical practitioners on the research team assisted with compiling keywords and/or phrases (see table 1). In order to take

**Table 1** Search terms for each topic

| Advance directives | OR | Mental capacity | AND | Suicidal behaviour |
|---|---|---|---|---|
| advance decisions<br>advance directives<br>advance statement<br>living will(s)<br>mental health directive<br>Ulysses contract(s)<br>psychiatric will(s)<br>antecedent decision/wish<br>pre-emptive suicide<br>antecedent refusal<br>resuscitation order<br>health care power of attorney | | mental competency<br>mental capacity | | suicide<br>attempted suicide<br>self-mutilation<br>self-harm<br>deliberate self-harm<br>parasuicide<br>self-injurious behaviour<br>drug overdose<br>self-immolation<br>self-poisoning<br>self-destructive behaviour<br>auto aggression<br>automutilation |

an inclusive approach and enable inclusion of any papers that involved discussion of management of advance decisions following 'suicidal behaviour' we included a variety of key search terms relating to non-accidental injury and suicide attempts. An electronic search of six databases (EMBASE, MEDLINE, PSYCHINFO, Social Policy and Practice, CINAHL and Medline) was conducted, as well as a full electronic search on WestLaw (an online library of UK legal information) using the following search terms: *advance decisions, advance directives* AND *wills, suicide*. Full search strategy for each database is supplied as supplementary information (online supplementary information 1). In addition, the reference sections of all included sources were consulted and authors' personal files were also searched to ensure that potentially eligible sources were not omitted. No study design, date or language restrictions were imposed.

Literature searches were conducted during the period April 2016–July 2018. The specific inclusion and exclusion criteria are detailed in table 2.

### Study selection

Titles and abstracts were screened, with a random sample of 10% of the articles independently screened by another researcher. Additional information was sought where there were any disagreements, which were then resolved through discussion. An acceptable concordance rate between the inclusion decisions was predefined as agreement on at least 90% of the articles, which was achieved for screening on title and abstract. Full text screening of the selected articles was conducted by two researchers independently, with full agreement being achieved at this stage.

**Table 2** Criteria for inclusion and exclusion

| Parameter | Inclusion criteria | Exclusion criteria |
|---|---|---|
| Patients | Patients over 18 years who present to hospital with advance decisions* (also include do not resuscitate orders, DNRs) following suicidal behaviour (including attempted suicide, deliberate self-harm, self-injurious behaviour, drug overdose, self-poisoning, self-destructive behaviour) with no existing chronic or terminal physical conditions. | Patients who present to hospital with advance decisions but with primary conditions which were not mental health related (eg, HIV/AIDS, chronic physical health conditions or disabilities, neurodegenerative diseases and/or specific patient groups (eg, mother/baby)). |
| Intervention | Medical management and/or medicolegal and/or ethical consultation/discussion. | Medical management of euthanasia, assisted suicide, end of life, wills/inheritance (ie, monetary or property issues). |
| Comparator | | |
| Outcomes | Adherence/non-adherence with advance decision, treatment, patient outcome (ie, death). | |
| Study design | Opinion and review articles, case studies, empirical studies/surveys. | Book reviews, responses to articles, conference abstracts. |

*Or other terms such as advance decisions, advance directives, advance statement, living will(s), mental health directive, Ulysses contract(s), psychiatric will(s), mental competency, mental capacity, healthcare power of attorney, antecedent decision/wish, pre-emptive suicide, antecedent refusal, resuscitation order or living will, advance directive, Ulysses contract.

## Data extraction and analysis

A preliminary analysis of the data was conducted.[18] Studies were from a range of disciplines (ie, general medical, psychiatry, ethical, legal) and involved reviews of clinical cases or fictional scenarios. It was deemed appropriate to conduct a narrative synthesis because this particular approach is useful when synthesising textual findings from diverse literatures.[18] Narrative synthesis was conducted in two phases: (1) a *textual synthesis* and (2) an adapted *thematic framework analysis.*[19]

First, the *textual synthesis* of the data was conducted by extracting key factual information from each study (country of origin, perspective/discipline, factual or fictional case study) and details of the case studies (age of patient, mental health disorders, nature of suicidal behaviour, resulting injuries/illness, hospital admittance, type of advance decision, when the advance decision was written, and whether patient was conscious, decision-making processes). The information was then summarised and tabulated to map the literature that cited the same clinical case. Information from cases only involving a factual case study (ie, a real clinical case) was extracted because we were interested in information about actual clinical cases, decision-making process and rationale for decisions made. Thus, information was not extracted from reports that discussed a hypothetical scenario for the textual synthesis. Data extraction and summarisation was completed independently by two researchers using a predetermined data extraction sheet.

Second, an adapted *thematic framework analysis* approach[19] was used to examine key themes discussed in the selected papers. This involved five stages: initial open coding, indexing, descriptive summaries, charting and tabulation and interpretation. *Initial open coding* generated three general categories representing the most discussed issues across the selected articles: (1) key issues with an advance decision relating to suicidal behaviour, (2) challenges in clinical decision-making for advance decisions relating to suicidal behaviour and (3) recommendations for practice. These three categories were used to index the data and as a framework to extract and summarise data. Extracted data were then used to form *descriptive summaries*. Indexing, extracting and summarising were conducted independently by two researchers. Resulting summaries were compared and discussions were held to clarify any differences. *Charting and tabulation* was conducted by charting the summaries by discipline. In order to explore similarities and differences between disciplines, we distinguished between 'General Medical' as papers written from a general medical practice or emergency services perspective; 'Psychiatry' as those written by clinical psychiatrists or from a psychiatry perspective, 'Nursing' as those written by practising nurses or research nurses, 'Bioethics' as those in ethics sections in journals or written by researchers in medical ethics, 'Ethics' as those in ethics journals or written by ethics researchers and 'Legal' as those written from a legal perspective and/or by a legal representative. *Interpretation*

of the data was conducted by thematic analysis of the summary charts to highlight the main recurrent and most important themes.[18] Two researchers conducted the thematic analysis independently and then discussed and finalised themes. Saturation of the themes was established when no further themes emerged and could not be further collapsed. 'Vote counting' was used to identify the frequency with which the themes appeared in the selected papers.[20] In the thematic framework analysis all selected studies were included; those involving a factual case and those involving a fictional case, because both involved discussions of concerns, challenges and rationale for decision-making relating to management of an advance decision following suicidal behaviour.

## Quality assessment

The papers mostly comprised accounts of clinical cases written by clinicians and ethical or legal experts. The methodology quality and synthesis of case series and case reports tool suggested by Murad and colleagues[21] was used to assess the quality of selected studies. Each study was assessed independently across four areas of potential bias: selection, ascertainment, causality and reporting. The tool consisted of five items each requiring a binary response to indicate whether the bias was likely. We considered the quality of the study good when all five criteria were fulfilled, moderate when four were fulfilled and poor when three or fewer were fulfilled. The methodological quality of included studies was assessed independently by two reviewers and discussions were held between them where there was disagreement. We also considered the reflexivity of the author/s, their expertise and how they were involved in the clinical case (eg, as a clinician or legal/ethics consultant). Authors of the papers reflected on the management of the clinical case, rationale for decision made and issues relating to advance decisions and suicidal behaviour more generally.

## Patient and public involvement

An expert-by-experience was a co-applicant on the National Institute of Health Research (NIHR) Programme Grant and actively contributed to the study design and objectives. Patient advisors, carers and clinicians evaluated the relevance and importance of the research questions for the advance decisions component of the grant and the systematic review. Our interim and final results were presented and evaluated by clinicians, academics, patients and carers. There was also patient input into our dissemination plan which includes dissemination to clinicians and the relevant patient community.

## RESULTS
### Systematic search
Results of the systematic search are displayed in figure 1. After duplicates were removed, the search returned 634 articles, of which 35 were retained after screening based

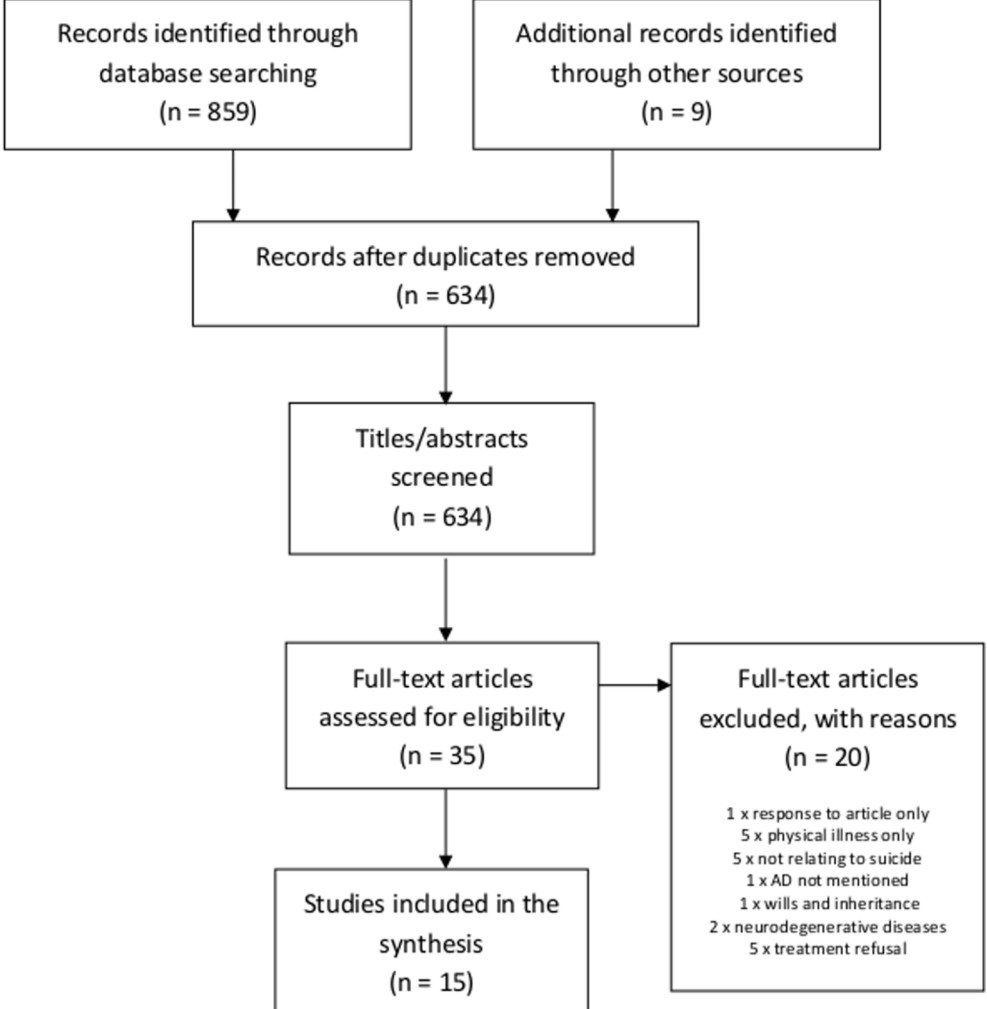

**Figure 1** Flow chart of results from initial search. AD, advance decision.

on title/abstract. Following full-text screening, 15 articles were retained for data extraction.

### Study characteristics

Descriptive information about the selected articles is displayed in table 3. Five of the selected articles were from the UK and the others were from the USA (n=7) or Australia (n=3). A total of six clinical cases were reviewed across the 15 articles (see table 3), as seven (47%) of the articles reported the same case (case A, a well-publicised case of a 26-year-old woman who died in the UK). Two of the clinical cases presented fictional scenarios.[2 22]

### Study quality assessment

All 15 studies were assessed for bias using the methodology quality and synthesis of case series and case reports tool suggested by Murad and colleagues.[21] Nine of the selected studies were deemed to have moderate methodological quality and six to have poor quality (see online supplementary information 2). The quality assessment is supplied as supplementary information (online supplementary information 2). None of the studies reported the representativeness or selection process relating to the case report, which impacted on the bias ratings. Although

case reports are considered to have increased risk of bias, they have profoundly influenced medical literature and advanced knowledge and their use in reviews is considered appropriate where no other higher level evidence is available.[21]

### Textual synthesis

#### Examination of clinical cases discussed in the selected articles

Specific information about clinical cases and decision-making is summarised and charted in table 4. We only included examination of the factual cases (n=6) in this part of the analysis, because we were interested in the types of real-world cases and decisions made, rather than an examination of a hypothetical scenario.

Patients discussed in the clinical cases varied in age, ranging from 26 to 86 years old. All patients were noted as having a diagnosis of depression, some were reported as also having diagnoses of post-traumatic stress disorder and personality disorders. The suicide methods used in the cases included self-poisoning (n=3), gunshot incidents (n=2) and hanging (n=1). All patients were found by other people, except one patient who called an ambulance because they did not want to die alone. Four of the

**Table 3** Description of selected studies

| Author | Date | Country | Perspective* | Fictional/factual case | Case reported† |
|---|---|---|---|---|---|
| Bryne[2] | 2002 | Australia | Nursing | Fictional | – |
| Callaghan and Ryan[26] | 2011 | Australia | Bioethics | Factual | A |
| Chalfin et al[25] | 2001 | USA, Philadelphia, New York, New Zealand | Emergency and acute medicine/bioethics | Factual | B |
| Cook et al[23] | 2010 | USA, Illinois | Psychiatry | Factual | C |
| Dresser[6] | 2010 | USA, New York | Legal | Factual | A |
| David et al[27] | 2010 | UK | Psychiatry | Factual | A |
| Frank[7] | 2013 | USA, Colorado | Legal | Factual | D |
| Kapur et al[8] | 2010 | UK | Psychiatry | Factual | E |
| Mitchell[22] | 2011 | USA, San Diego | Ethical | Fictional | – |
| Muzaffer[28] | 2011 | UK | Psychiatry | Factual | A |
| Richardson[29] | 2013 | UK | Legal | Factual | A |
| Ryan and Callaghan[30] | 2010 | Australia | Psychiatry | Factual | A |
| Sontheimer[24] | 2008 | USA, Springfield | Bioethics | Factual | E |
| Szawarski[31] | 2013 | UK | Bioethics | Factual | A |
| Volpe et al[32] | 2012 | USA, New York | Bioethics | Factual | F |

*Where the perspective is not clearly stated, this has been derived from the author(s) background and professional experience.
†For specific details about each case, see table 4. Note: fictional cases have not been given a case report ID.

patients were reported to have died; the outcome in one case was not specified.

Treatment was provided in only one of the clinical scenarios.[23] In this case, the patient was a psychiatric inpatient and the advance decision was considered part of the suicide attempt, so the patient's treatment refusal specified in the advance decision document was not adhered to.

The rationale for non-treatment in the clinical cases where the patient died varied and was summarised into the following three reasons:

► Advance decision was followed as a *legally-binding document* after checks showed the information was clear and specific, patient was informed of treatment options, had mental capacity at the time of writing and family were in agreement with the decision for non-treatment (n=1).[8 24]

► Physical injuries were severe resulting in *poor prognosis* for the patient and the treatment refusal in the advance decision was used as evidence that the patient would not wish to survive with a life-threatening or severely disabling condition. Where possible, families were also consulted (n=2).[7 25]

► *Verbal treatment refusal* was used as the basis for the treatment decision, rather than the advance decision, because the patient was conscious and had mental capacity. Consultation with family was not reported in this case. (n=1).[6 26–31]

The decision-making process was reported to take considerable time and legal and/or ethical consultation took place in all the reported clinical cases.

Differences in opinions about clinical management and decision-making between ED clinicians and psychiatric consultants were reported in some of the clinical cases.[23 25] In those cases, ED clinicians gave more weight to the advance decision, suggesting it should be adhered to as a legally binding document and the patient remain untreated. In contrast psychiatrists viewed suicide as a consequence of a distressed state and expressed a preference to avoid adherence with the advance decision and treat the patient. Where such conflict arose this was resolved through consultation with the hospital legal team and/or ethics committee.

### Thematic analysis
Five themes arose from the thematic analysis and are presented with their corresponding subthemes and votecounts in table 5. We included accounts of fictional cases in the thematic analysis because here we were interested in opinions, views and perspectives of authors.

### Themes
Tension between patient autonomy and protecting a vulnerable person
*Professional dilemma: promoting patient autonomy versus providing appropriate care*
The management of an advance decision in the context of suicidal behaviour was particularly challenging because it went against healthcare professionals' training to preserve life (ie, adherence to the advance decision could result in the death of the patient while they could recover if they received treatment for their physical condition). This

**Table 4** Description of clinical cases discussed in selected studies

| Case | Reference | Age | Mental health conditions | Nature of SA | Resulting injuries/illness | Hospital admittance | Nature of the AD | When written? | Patient conscious? | Decision-making process | Rationale for decision | Outcome |
|---|---|---|---|---|---|---|---|---|---|---|---|---|
| A | 6 26–31 | 26 | Depression generalised anxiety disorder, PTSD, BPD | Self-poisoning (antifreeze) | Not stated | Presented herself at hospital | Letter | 3 days prior | Yes | Medical staff discussed the patient's mental capacity and sought legal advice. | The patient's wishes were clear in the letter but the patient was conscious, judged to have capacity and refusing treatment. | Death |
| B | 25 | 46 | Severe depression | Gunshot to face | Pain and severe facial injury | Gunshot reported by neighbours | Suicide note | Not stated | Yes (not coherent) | The attending physicians thought life-support should be removed as the patient's 'will' was clear and authoritative. The psychiatrist thought suicide was pathological and the condition was treatable so the patient should be treated. Clinicians consulted widely and sought legal advice. | The suicide note was accepted as a living will. The patient had a desire to die due to psychological pain. The suicide attempt left the patient in a severely disabled state. | Death |
| C | 23 | 57 | Depression generalised anxiety disorder, PTSD, BPD | Self-poisoning (opiates) | Respiratory distress | Psychiatric inpatient | DNR | Prior to inpatient admittance | Not stated | There was conflict between clinicians; the psychiatrist argued that the DNR should not be followed because it was a suicide attempt. The legal/ethics committee was consulted who supported continued treatment. | DNR considered an effort to prepare for a suicide attempt and should not be honoured. | Survived and regretted the suicide attempt. |

Continued

**Table 4** Continued

| Case | Reference | Age | Mental health conditions | Nature of SA | Resulting injuries/illness | Hospital admittance | Nature of the AD | When written? | Patient conscious? | Decision-making process | Rationale for decision | Outcome |
|------|-----------|-----|--------------------------|--------------|-----------------------------|---------------------|------------------|---------------|--------------------|--------------------------|------------------------|---------|
| D | 7 | 35 | Depression and drug abuse | Hanging | Brain injury | Found by family | AD | Not stated | No | There were concerns that adherence to the AD would result in the patient's death. Clinicians sought legal advice. | The patient had poor prognosis and the family gave consent for clinicians to stop treatment. | Death |
| E | 8 24 | 52 | Depression generalised anxiety disorder, PTSD, BPD | Self-poisoning (insulin) | Coma | Found at home | AD | 2 years prior | No | The AD mentioned no treatment for a terminal condition. The patient was not in a terminal condition and there were concerns that injury was the result of a suicide attempt and whether the AD should be adhered to in a suicidal context. Approached family and held an ethics committee consultation. | The patient's wishes were judged to be clear, the patient was considered to be informed about treatment options and had mental capacity at the time of writing the AD and the family were in agreement. | Death |
| F | 32 | 86 | Not stated | Gunshot to chest | Damage to pancreas and colon | Not stated | AD | Not stated | Yes (not always coherent) | Medical team argue that the nature in which the physical condition was caused (ie, suicidal behaviour) should impact on treatment. | Not stated | Not stated |

For details about articles, see table 3.
AD, advance directive; BPD, borderline personality disorder; DNR, do not resuscitate; PTSD, post-traumatic stress disorder; SA, suicide attempt.

**Table 5** Themes from the selected articles

| Theme | Subthemes | Theme descriptor | Perspectives | References | Count (%) |
|---|---|---|---|---|---|
| Tension between patient autonomy and protecting a vulnerable person. | *Professional dilemma: promoting patient autonomy versus providing appropriate care.*<br>*Societal expectation to protect vulnerable person and prevent suicide.* | Tension between acting in accordance with patients' wishes for their medical treatment while promoting their best interests presented clinicians with a professional ethical dilemma. Clinicians also had a personal ethical dilemma, as there is societal pressure to protect vulnerable people and prevent suicide. | Psychiatry, bioethics, legal. | 7 22 24 27 29 | 5 (33) |
| Appropriateness of advance decisions for suicidal behaviour. | *Mental health symptoms and suicidal ideation fluctuate.*<br>*Advance decisions for mental and physical health conditions—are they the same?* | There were questions about whether an advance decision 'fits' in relation to suicide without an existing physical illness because mental state, mental health and suicide ideation fluctuate. Such scenarios are different from decisions made about treatment for a chronic or terminal physical condition. | Medical, psychiatry, bioethics, legal. | 2 6–8 23–25 27 29–32 | 12 (80) |
| Uncertainty about the application of legislation. | *Confusion and anxiety about litigation.*<br>*Advance decisions are about more than a simple assessment of capacity.* | Legislation around advance decisions was seen as confusing and there was anxiety about ligation. It was noted that mental capacity legislation overlapped with mental health legislation and policy. There were concerns that relying on a capacity decision was not sufficient and the authenticity of the advance decision needed to be considered. | Medical, psychiatry, bioethics, legal. | 2 8 22–29 31 | 11 (73) |
| The length of time needed to consider all the evidence versus rapid decision-making for treatment. | *Need to fully consider the totality of evidence.*<br>*Increased gravity of the clinical decision.* | Clinical decisions were considered to be complex, involving an assessment of mental capacity, verification of the advance decision and consideration of contextual factors. Therefore sufficient time was needed in which to consider all of the evidence. | Medical, psychiatry, bioethics, legal. | 2 8 25–27 | 5 (33) |
| Importance of seeking support and sharing the decision. | *Drawing up an advance decision as a collaborative process.*<br>*Shared decision-making.* | Sharing the decision-making and seeking support, both at the time of writing the advance decision and when treating the patient, was viewed as important. | Medical, psychiatry, bioethics, legal. | 2 7 24–28 30 31 | 9 (60) |

presented clinicians with a dilemma between promoting patients' autonomy by observing their wishes stated in the advance decision and by providing care that was considered in their best interests (eg, promoting life).[7 23 26 28 30]

### Societal expectation to protect vulnerable person and prevent suicide

Authors also raised the issue that clinicians not only had a professional interest in protecting a vulnerable person, but there was also a societal expectation that suicide should be prevented.[23 25 30]

> While the right to autonomy is strong, in some circumstances there may be competing rights and interests that are sufficient to override a competent decision to refuse treatment. These may include the state's interests in preventing suicide.[30]

The challenge to clinicians was highlighted by an acknowledgement from some authors that adherence to the advance decision in this context was emotive and would feel like assisting suicide.[24 30]

### Appropriateness of advance decisions for suicidal behaviour
*Mental health symptoms and suicidal ideation fluctuate*
Concerns were expressed about whether an advance decision should apply in the context of suicidal behaviour because of the patients' distressed state, the potential for suicidal ideation to fluctuate and for treatment preferences to change in the future.[7 8 31 32]

> The compelling notion that people will change their minds contradicts the primacy of patient autonomy in the consideration of suicide. This is what distinguishes an impulsive suicide attempt from other informed choices to obtain or refuse medical treatment by patients.[7]

Authors from a psychiatric perspective, in particular, viewed suicidal behaviour as a symptom of a mental health disorder that was potentially treatable with psychiatric care.[25] They also expressed concerns about the capacity of a distressed suicidal person to fully comprehend their decision and consider all treatment options available to them.[2 24 25 32] Therefore, it was suggested by some authors that a higher level of mental capacity may be required at the time of writing the advance decision for clinicians to be confident in following it.[8] However, other authors argued that the advance decision should be considered as part of the suicide attempt and as evidence of distressed/disordered thinking,[8 23 27 28] rather than independently of the attempt and the treatment refusal in the advance decision document should not be adhered to.

### Advance decisions for mental and physical health conditions—are they the same?
The difference between an advance decision for suicidal behaviour and for a physical condition was highlighted across the selected papers.[6 32] Authors from a legal perspective highlighted that the primary aim of an advance decision relating to a suicide attempt is to end life, whereas an advance decision for a chronic or terminal illness is often concerned with managing pain and avoiding prolonged suffering.[6]

There was also debate about the extent to which mental suffering legitimised suicide.[32] Authors from an ethical perspective argued that, typically, healthcare services may be more sympathetic to 'end-of-life' decisions relating to terminal physical health conditions than mental health conditions, thus mental health patients do not receive the same palliative care options as patients without mental health diagnoses.[24] There was some discussion that it should not be assumed that psychiatric pain is more tolerable than physical pain and that both should be considered as having a similar influence on the patient.[24 25]

### Uncertainty about the application of legislation
*Confusion and anxiety about litigation*
Authors from general medical and psychiatry perspectives expressed confusion about legislation and anxiety about litigation,[2 23 30] with one stating that the advance decision document needed to be '*watertight*' to be considered.[25] Authors recommended that clear hospital policies be developed for advance decisions in this particular context to overcome the confusion and anxiety about ligation.[23]

> In addition to the clinical demands associated with treating a patient with a life-threatening condition, clinicians must do their best to ascertain the patient's capacity for his or her apparent decision, consider the correct ethical course, and navigate through uncharted legal waters.[7]

Authors from the UK and Australia highlighted the difficulties in implementing both mental health and mental capacity legislation when managing advance decisions relating to suicidal behaviour.[27 29–31] Clinicians needed to consider whether someone who had attempted suicide was suffering with a mental health condition, for which they should be treated against their will. They also needed to judge whether the person had the capacity to make a decision about their treatment and, if so, that the advance decision could apply following verification checks. Some suggested that application of each legislation model (ie, mental health or mental capacity), in isolation of the other, could result in different outcomes for the patient.[6] Some authors suggested that the difficulty with balancing mental capacity legislation and mental health legislation could be resolved by developing a single legislation that combines both.[8 27]

### Advance decisions are about more than a simple assessment of capacity
A reliance on judging a person's capacity to make a decision in the context of suicidal behaviour was discussed in detail.[8 22 24] The capacity assessment was discussed in relation to when the patient was involved in advance care planning and making the decision to write an advance decision to refuse treatment.[8] Capacity assessment was

also discussed in relation to clinicians in an emergency situation, treating a person who is considered to have capacity to verbally refuse or accept treatment. In this scenario the advance decision can be ignored. While this is an important part of some legislation, particularly in the UK, it was suggested that an assessment of capacity should be supplemented with a judgement of the authenticity and durability of the patient's decision (ie, if the decision had been consistent over time).[22 26] Authors from a psychiatric perspective, in particular, suggested that advance decisions should be regularly reviewed to ensure that they were up-to-date and continued to reflect the patient's desires and preferences.[26–28]

### The length of time needed to consider all the evidence versus rapid decision-making for treatment
#### Need to fully consider the totality of evidence
Some authors suggested that the increased length of time taken in this particular context arose from the need to consider contextual factors for the suicidal behaviour,[2 22 25] the patient's mental health background[27] and the reason for their decision, alongside the usual validation checks and judgement of the presence of mental capacity at the time of making the advance decision. It was also argued that clinicians should take into account wider factors that may have not been present when the person first wrote the advance decision, such as changes in evidence for a particular treatment or scientific advances offering new treatment options that may influence the patient's decision.[22]

However, authors highlighted difficulties with gaining access to such evidence, particularly in emergency situations, further adding to the time taken to make a decision.[31] It was noted that advance decisions were often too specific (eg, related to a specific illness or injury) or too general (eg, a general refusal of treatment, rather than refusal of a specific treatment), resulting in ambiguity as to the best course of action for the patient and time consuming investigation.[2 25 28] Some authors highlighted that advance decisions were not useful in emergency settings when rapid decision-making was required[2] but may be appropriate for patients to express refusals of on-going psychiatric treatment (eg, electroconvulsive therapy).

#### Increased gravity of the clinical decision
Authors argued that the gravity of the clinical decision was increased in this context because the patient could die if the advance decision was adhered to when recovery from mental ill health may be possible.[6 25] Authors suggested that validation checks in this context may need to be more thorough and authors from a legal perspective argued that, because of the increased gravity of the clinical decision, physicians should seek a consensus about clinical management, while providing life-sustaining treatment, creating a time-consuming situation.[7 31]

### Importance of seeking support and sharing the decision
#### Drawing up an advance decision as a collaborative process
Some authors argued that when writing an advance decision, patients should be supported by a healthcare professional

to consider all possible treatment options.[2 22 23 27 29] It was suggested that evidence of mental capacity at the time of writing the advance decision should be provided (eg, verified and signed by the healthcare professional) which could help with clinical decision-making at a later stage.[22] Authors from all the perspectives stressed the importance of also consulting with a physician at the time of writing the advance decision to ensure that it is both specific and general enough to be helpful and informative in a given medical scenario.[23 27]

#### Shared decision-making
All authors discussed the need for multiagency decision-making in relation to the management of advance decisions in the context of suicidal behaviour.[7 27 28] Suggestions included that clinicians should consult widely, make use of psychiatric expertise, review the patient's psychiatric history and background and seek legal and/or ethical consultation when considering treatment decisions.

## DISCUSSION
### Summary of the findings
A comprehensive systematic review of studies examining the management of advance decisions to refuse treatment following suicidal behaviour was conducted. The findings show a paucity of studies in this specific area. Fifteen relevant studies were identified, of which all were reports of clinical cases. With the exception of two papers that noted fictional clinical cases, the others reported on six real clinical cases. Despite having no language or country restrictions to the search, all the studies were from the USA, Australia or UK which have similar legislation relating to advance care planning and advance decisions to refuse treatment.[2]

There were inconsistent views on practice and rationales for the management of advance decisions. Treatment was provided in only one clinical case, where the patient was a psychiatric inpatient and the advance decision was considered part of the suicide attempt.[23] In this case the patient survived and later regretted the suicide attempt. In the other clinical cases, treatment was not provided, but rationale for non-treatment differed. Rationale for treatment varied from viewing the advance decision as legally binding[8 24] to using the advance decision as an aide to understand the patients' treatment preferences when there was a poor prognosis or a resulting severely disabling condition.[7 25]

Conflict between clinicians was reported in some of the cases.[23 25] In the studies where there were conflicts, there were differences in opinions on treatment between ED clinicians and psychiatrists. Consultations with mental healthcare staff were typically sought when a patient presented with an advance decision following suicidal behaviour. Psychiatrists tended to stress the treatable nature of a mental health condition and that the suicidal behaviour was part of the mental health disorder. In contrast, ED clinicians argued that the advance decision

document was legally binding and expressed anxieties about litigation. These differences in opinion about treatment were overcome through consultations with legal and ethical representatives.

The appropriateness of advance decisions with suicidal behaviour was questioned for two reasons. First, suicide ideation was considered to fluctuate and people could change their mind about their desire to die.[7 8 31 32] Although suicide has been linked to impulsivity,[33 34] studies show that not all suicides are impulsive.[35] However, recent studies using ecological momentary assessment have shown that suicide ideation varies over short periods of time (ie, there are changes between hours and days)[36] and follow-up studies with suicide survivors tend to acknowledge that they regret the suicide attempt.[37] Second, outcomes for treatment refusal following suicidal behaviour were noted to be potentially different to those for a terminal physical health condition (ie, the patient could die when there is potential for recovery in the future).[6 32]

Authors discussed concerns that management of advance decisions following suicidal behaviour may need to be different because they are a unique clinical presentation. Similar to findings in this review, anxieties and confusion about legislation relating to advance decisions is also found in studies examining end-of-life care.[38] However, what differs is opinions about adherence to the advance decision to refuse treatment for chronic or terminal conditions and sympathy for assisted suicide in end-of-life care. Healthcare workers report support for assisted suicide relating to end-of-life care[39] and frustrations with continuing life-sustaining treatment where withdrawing treatment might be considered in the best interest of the patient when they have a life-threatening condition.[23 40] Those findings indicate quite a contrast with opinions in this review where the focus was on management of advance decisions following suicidal behaviour and an expression of sympathy with the decision was not found. It will be important in future research to examine these differences further by contrasting views on management of advance decisions to refuse treatment following suicidal behaviour for patients with chronic and/or terminal physical conditions and patients without chronic or terminal physical conditions.

Management of the advance decision was difficult both emotionally and ethically for some clinicians because it challenged their professional training and their desire to protect vulnerable patients from suicide. The competing pressures of respecting a patient's right to autonomy while protecting them from the effects of mental disorder found in the current study is a commonly reported dilemma.[41] There is evidence from the present study that support for the right to autonomy may be more dominant in clinicians from emergency medicine disciplines, with those from a psychiatric background prioritising prevention of suicide. A 'middle ground' between these views may help to provide guidance for clinicians. For example, in English law, courts

have acknowledged that while some suicidal individuals may have capacity, the overwhelming likelihood is that capacity is impaired to at least some degree.[41] Suicidal ideation has been associated with disordered and impulsive decision-making[33 34] and evidence indicates that most mental health patients presenting to EDs are judged as not having capacity to make a treatment decision.[12] Therefore a higher degree of certainty should be required when assessing capacity with suicidal behaviour and clinicians should err on the side of caution.[8] Another potential resolution to this dilemma, particularly in emergency scenarios, may be to provide 'temporary intervention' to allow time for individuals to be assessed and treatment options to be discussed.[41]

An added pressure for clinicians in the management of advance decisions following suicidal behaviour was that they felt there was a societal expectation that suicide should be prevented. Adhering to the advance decision made by the patient by not treating them, not only was seen to go against their professional training to protect the patient, but it was viewed that this may be considered from a society perspective as unacceptable. The dilemma here is that a clinical decision of non-treatment and adherence with the advance decision might be accepted legally, but not socially. Concerns were expressed that this particular presentation of an advance decision met conditions that warranted overriding patients' autonomy because non-adherence with the advance decisions results in prevention of suicide, maintenance of the integrity of the medical professional and preservation of life.[25]

### Recommendations for practice

Decisions made about advance decisions in the context of suicidal behaviour should be made in full consultation with psychiatric teams and with relevant legal and/or ethical advisers. The results also highlight the importance of allocating sufficient time to consider contextual evidence relating to the suicidal behaviour, the authenticity of the treatment decision and verification of the documentation/decision. Given the gravity and emotive nature of a decision in this context, emergency healthcare workers may need increased support and supervision for such incidents.

Findings indicate that it may be helpful, in this particular context, for an advance decision to be written in consultation with a professional healthcare worker and the patient's family. This practice would also ensure that the patient is supported to consider all treatment options, that the advance decision is specific and detailed enough to be useful in an emergency situation and that patients' capacity at the time of writing the advance decision can be assessed and verified. The advance decision should be regularly reviewed and updated to ensure that it reflects the patient's current treatment decisions.

## Strengths and limitations

A strength of this review is that a broad range of articles from different disciplines were included, thus increasing the generalisability of results. However, there were some potential biases in the literature. First, there was a paucity of evidence: only six clinical cases were reported across the selected articles. There was also a risk of bias from the studies themselves, given that they were reviews of single clinical cases. Second, the articles were focused on the USA, UK and Australia, so may have resulted in bias relating to the specific legislation/ethics of those countries. There may be different views on this topic and its management in countries with different implementation of legislation, so it will be important for future research to compare findings internationally across a wider range of countries.[42–44] Third, as with any syntheses of qualitative data there was potential for bias to be introduced by the research team at the stages of study identification, data extraction and synthesis. This was minimised in the current study by having two researchers carry out these tasks independently and cross-check the findings.

## Future directions

Empirical studies, such as interviews and focus groups with clinicians and patients and/or a national clinical survey are important future priorities. Given that the presentation of an advance decision following suicidal behaviour is rare, case reports are likely to continue to be important sources of information in the future and authors should be mindful to ensure that case reports include details about how information about the case were obtained and how representative it is of other cases in this area. Research examining the prevalence of advance decisions relating to suicidal behaviour could shed light on the frequency of such presentations. Suitable platforms for storing advance decisions could also be explored. For example, some have suggested a web application ('app') could better reflect the dynamic nature of treatment refusal[45] and make updating and reassessment easier.

## CONCLUSION

Current literature on the management of advance decisions and suicidal behaviour centres on detailed accounts of clinical cases and demonstrates variability in practice and the rationale behind clinical decisions. Challenges in managing advance decisions specific to suicidal behaviour were evident, and there was some debate about whether advance decisions in the context of suicidal behaviour were appropriate in their current form. Taking time to consider all the evidence when making a decision, consulting fully with mental health clinicians and seeking legal and/or ethical advice may help with some of these challenges. The support of a relevant healthcare professional at the time of writing the advance decision may also be useful.

### Author affiliations

[1]Centre for Mental Health and Safety, University of Manchester, Manchester, UK
[2]NHIR Greater Manchester Patient Safety Translational Research Centre, University of Manchester, UK
[3]Centre for Ethics in Medicine, University of Bristol, Bristol, UK
[4]Centre for Suicide Research, University Department of Psychiatry, Warneford Hospital, Oxford, UK
[5]Department of Population Health Sciences, University of Bristol, Bristol, UK
[6]School of Law, University of Manchester, Manchester, UK
[7]Emergency Department, Manchester Royal Infirmary, Manchester, UK
[8]Greater Manchester Mental Health NHS Foundation Trust, Chorlton House, Manchester, UK

**Acknowledgements** We would like to thank Rosie Davies, Research Fellow in Patient and Public Involvement at the University of Bristol, for her input into the study.

**Contributors** All authors made substantial contributions to the study. RN and LQ conducted the initial scoping search. RN designed the review and data extraction/analysis with input from NK and SS. RN and SS screened and reviewed the articles and performed data extraction/analysis, interpreted the results and wrote the first draft. NK, JC, RH, KH, DG, NA and KMJ reviewed the initial draft, contributed to subsequent drafts and approved the final version. All authors take responsibility for the integrity of the data analysis. NK is the guarantor of the study.

**Funding** This paper presents independent research funded by the National Institute of Health Research (NIHR) under its Programme Grants for Applied Research Programme (Grant Reference Number RP-PG-0610-10026).

**Disclaimer** The views expressed are those of the authors and not necessarily those of the NHS, the National Institute of Health Research or the Department of Health.

**Competing interests** DG, KH and NK are members of the Department of Health's (England) National Suicide Prevention Advisory Group. NK chaired the NICE guideline development group for the longer-term management of self-harm and the NICE Topic Expert Group (which developed the quality standards for self-harm services). He is currently chair of the updated NICE guideline for Depression. KH and DG are NIHR Senior Investigators. KH is also supported by the Oxford Health NHS Foundation Trust and NK by the Greater Manchester Mental Health NHS Foundation Trust.

**Patient consent for publication** Not required.

**Provenance and peer review** Not commissioned; externally peer reviewed.

**Data sharing statement** No additional data are available.

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
