## [Reviewer comments · BMJ Open]

ARTICLE DETAILS

TITLE (PROVISIONAL)	The management of patients with an advance decision and suicidal behaviour: A systematic review
AUTHORS	Nowland (Harris), Rebecca; Steeg, Sarah; Quinlivan, Leah; Cooper, Jayne; Huxtable, Richard; Hawton, Keith; Gunnell, DJ; Allen, Neil; Mackway-Jones, Kevin; Kapur, Navneet

VERSION 1 – REVIEW

REVIEWER	Karolina Krysinska Centre for Mental Health, Melbourne School of Population and Global Health, University of Melbourne, Australia
REVIEW RETURNED	08-Jun-2018

GENERAL COMMENTS	Thank you for an opportunity to comment on a manuscript presenting a systematic literature review on the management of patients with an advance decision and suicidal behaviour. The topic of advance care planning is of much importance for clinical practice and suicide prevention. It is also extremely complex as can be evidenced by the reported results of the literature review and problems with the manuscript listed below. My major problem with the text (and the literature review) is lack of clarity. This includes very vague definition of “suicidal behaviour” (or a very vague phrase used in the text “in the context of suicidal behaviour”). The Introduction should clarify the context of the study - country and legal arrangements which are the background of the study, i.e., UK. I believe authors from other countries could approach the topic differently. For instance, I am not sure how to understand a very broad statement “Advance care planning and advance decisions for psychiatric care are becoming more common (WHERE?) and have some potential benefits, including enhancing patient autonomy and engagement (IN THE CONTEXT OF?), promoting adherence to treatment plans (SUCH AS?), improving continuity of care with fewer psychiatric admissions (IN WHICH COUNTRIES AND SETTINGS?), reducing the use of social workers’ time and lower levels of violent acts (Campbell & Kisley, 2012; Swanson et al., 2000). However, concerns have been raised (BY WHOM?) about clinical management (BY WHOM?) of advance decisions in the particular context of suicidal behaviour (MEANING?) (e.g. Dresser, 2010; Frank, 2013).” Also, it is not clear what is meant by “that decision” (page 7, line 6) and “advance decisions relating to suicidal behaviour” (page 7, line 23-24). The Rationale and the Aim of the review are not clear, and should be re-phrased. Please, provide a rationale for excluding “primary conditions which were not mental health related” and define “pre-existing mental
--

	health issues" (Table 2). Please, specify "range of disciplines" (page 10, line 27) and "factual information" (page 10, line 38). Please, note that patients/public could have been involved in deciding whether the review topic could be informative from their perspective. Please, provide a rationale for excluding hypothetical scenarios from some parts of the review (textual analysis?) and including such papers in other analyses (thematic analysis). The perspectives included in the thematic analysis (such as general medical and psychiatry) have not been specified in a systematic manner. It is not clear which themes identified in the review apply to all cases of clinical management of patients with advance care planning and which apply specifically to patients engaging in suicidal behaviour (e.g., promoting patient autonomy vs providing appropriate care). It is also not clear how the very complex "capacity assessment" issue should be applied to the moment an individual makes an advance care plan vs the moment of engaging in suicidal behaviour vs expectations of care after a suicide attempt. Given the current limitations of the manuscript regarding its aims and data analysis and presentation of results, I am not able to comment on Discussion of results, recommendations, future directions and conclusion.
--	---

REVIEWER	Jo Robinson Orygen, The National Centre of Excellence in Youth Mental Health, Australia.
REVIEW RETURNED	11-Jun-2018

GENERAL COMMENTS	This systematic review examining the management of patients with an advance decision and suicidal behaviour deals with an important, and ethically challenging topic. The review was well conducted and the manuscript is well written. I only have a few minor comments detailed below. I note that the search was conducted almost a year ago and suggest that it is updated before final submission. In the Introduction the authors discuss the variation in legislation across nationalities and cultures. The search was not restricted to English language articles but all the studies retrieved were conducted in the UK, the USA or Australia. I suspect there would be differing views on this topic and its management in different countries, in particular countries with different implementation around this legislation, and I think this is worth coming back to briefly in the discussion. It might be helpful to the reader to add references to the Results section so that it is easy to see which studies the authors are referring to in each point being made. I realise this information is in the tables but as a reader it may be easier if this information was also in the text. It would also give a quick reference to how many of the studies supported each point. There are a couple of points made in the Results that it would be good to dig into a bit deeper in the Discussion if possible. An example being where there are similarities and differences in end of life decisions between suicide and other physical health conditions. I think this is of real interest and the challenges of
--

	balancing individual autonomy with the professional views regarding the fluctuating nature of suicidality and the treatability of mental ill-health is worth additional discussion. I also think the statement about the 'societal expectation that suicide should be prevented' is worth exploration in the Discussion if possible. The preventability of suicide is something that we are at pains to emphasise as a sector for a range of reasons, but perhaps there is another side to that that is worthy of some consideration. Also in the Results section, I am not sure the term 'technological advances' (p21) is the right term – perhaps 'growing evidence base' or 'scientific advances' might be better. In the next paragraph where you state that sometimes advance decisions were reported to be 'too specific or too general' - I think an example or two may help the reader here i.e. what is too specific or too general when it comes to a document like this? At the beginning of the Discussion it may be helpful to remind the reader of how many studies were retrieved etc. I also think referencing the studies as you discuss them would be helpful throughout this section as well as in the Results. It is noted that there was evidence of differences in attitudes of clinicians from different backgrounds and I think it would be worth ensuring that this comes across clearly in the Results section. Perhaps also provide some additional explanation as to what is meant in the Allen reference that in most cases capacity will be diminished thus adding to the complexity of advance decisions in this population. This to me seems to be very important and some additional discussion/ debate would be of interest here. It would be interesting to get a sense of how the findings reported here compare to those in studies or reviews of this type of practice/legislation in the physical health sector, and if differences exist what are the implications of this for mental health patients and staff? Overall I think this is a well-conducted study and a well-written manuscript. My the comments above really reflect that this is an important and ethically challenging topic that just warrants a bit more in-depth discussion if this is possible. As the authors note this is likely to be an increasing issue for clinicians to have to deal with, so the more detail and/or advice (including in terms of recommendations for practice and policy) that can be provided here the better. I hope these comments are helpful. Best wishes.
--	--

REVIEWER	Marianne Wyder Australia
REVIEW RETURNED	26-Jun-2018

GENERAL COMMENTS	I thought this was a very interesting article in an important area. I did however find that the introduction lacked a bit of depth and it was only when I read the results that the importance of the topic became even clearer. It would be good to include a section in the introduction why this topic is important and why we need to think about it.
---

	I was also unclear about the definition of advance directives and when they were used. it would be good to provide some clarity around this. In particular in what context these would be considered (i.e after an attempt? or any other situations). It would also be good to include the various references in the actual result section. I also wonder if families should be included in the development of guidelines.
--	--

VERSION 1 – AUTHOR RESPONSE

Notes to Editor/reviewer

Editorial requests:	
Please revise the 'Article Summary' section of your manuscript (after the abstract). This section should contain only a 'Strengths and limitations' section which should contain five short bullet points, no longer than one sentence each, that relate specifically to the methods	We have now removed the other sections and only have a strengths and limitations section. We have added an addition bullet point, so there are now five.
Along with your revised manuscript, please include a copy of the ENTREQ checklist, for reporting of synthesis of qualitative research, indicating the page/line numbers of your manuscript where the relevant	We have now included an ENTREQ checklist. This includes page numbers were certain aspects of the checklist are found in the manuscript.
The full search strategy for at least one database needs to be provided as a supplementary file and referred to in the methods section of the manuscript.	We have added text on page 9 in the search strategy and data sources section that states that the full search strategy for each database is supplied as a supplementary file. We have added an additional file (supplementary information) including the search strategy for each of the databases.
The quality of study reporting in relation to your search needs improving. For example, were databases searched from inception? Where any language restrictions applied? We recommend using the PICOS format for the inclusion criteria.	We have added the following text on page 9 in the search strategy and data sources section. “An initial scoping of the literature was conducted at inception of the study and the findings were used to inform the search strategy” We have stated in the manuscript that no language restrictions were applied on page 9 in the search strategy and data sources section. We have changed Table 2 now so it follows the PICOS format for inclusion and exclusion criteria.

Associate Editors comments to Author:	
This is an important and complex research topic: patients with an advance care plan and suicidal behaviour Can they clarify the dates of the search – and if over a year ago can they update it.	We have updated the search which generated one potential source which did not meet our inclusion criteria because it was a conference abstract. The dates of searches in the text have been changed to July 2018.
Can they provide a search string for at least one database	We have added text on page 9 in the search strategy and data sources section that states that the full search strategy for each database is supplied as a supplementary file. We have added an additional file including the search strategy for each of the databases.
Can they report their assessment of the study quality.	We now report explicitly on the assessment of study quality and include a table with this information included (Supplementary information 2).
The Discussion section is rather brief.	The discussion has now been increased in length based on reviewer 2's comments to include additional discussions and increase the depth of discussion of some points of interest.
Reviewer(s)' Comments to Author:	
Reviewer 1	
Thank you for an opportunity to comment on a manuscript presenting a systematic literature review on the management of patients with an advance decision and suicidal behaviour. The topic of advance care planning is of much importance for clinical practice and suicide prevention. It is also extremely complex as can be evidenced by the reported results of the literature review and problems with the manuscript listed below. My major problem with the text (and the literature review) is lack of clarity. This includes very vague definition of “suicidal behaviour” (or	

a very vague phrase used in the text “in the context of suicidal behaviour”).	We have addressed this by providing a comment at the end of the introduction to explain that we refer to both self-harm and suicide attempts in the paper when we state “suicidal behaviour” in order to be inclusive. “The terminology for suicidal behaviour varies internationally. Some clinicians/researchers distinguish between suicide attempts and non-suicidal self-injury, while others prefer the broad term of self-harm to denote behaviours across the spectrum.^{1,9} We took an inclusive approach to ensure we captured relevant studies, so in this review we refer to “suicidal behaviour” as behaviours including all self-harming behaviour and suicide attempts.” We have also changed the text in a number of places in the introduction to make the context clearer, e.g. on page 7 instead of management of suicidal behaviour it now reads “The management of self-harm and suicide attempts”. We have removed any statements of “context of suicidal behaviour” and have been more explicit about the specific context we are referring to. We feel that in this version of the paper the context is much clearer.
The Introduction should clarify the context of the study - country and legal arrangements which are the background of the study, i.e., UK.	We have included discussion of the legal perspective from a number of countries and highlighted the differences in the first couple of paragraphs of the document. We have added a few sentences after the aim in the last paragraph of the introduction to highlight to the reader that the study was conducted in the UK. We have the following comment in the discussion “the articles were focussed on the US, UK and Australia, so may have resulted in bias relating to the specific legislation/ethics of those countries” on page 23 to highlight to the reader the specific background to the findings.
I believe authors from other countries could approach the topic differently. For instance, I am not sure how to understand a very broad statement “Advance care planning and advance decisions for psychiatric care are becoming more common? (WHERE) and have some potential benefits, including enhancing patient autonomy and engagement (IN THE CONTEXT	We thank the reviewer for pointing this out. We have amended this text to make it more precise and to improve clarity. It now reads: Advance care planning for psychiatric care is becoming more common in a number of

OF?), promoting adherence to treatment plans (SUCH AS?), improving continuity of care with fewer psychiatric admissions (IN WHICH COUNTRIES AND SETTINGS?), reducing the use of social workers' time and lower levels of violent acts (Campbell & Kisley, 2012; Swanson et al., 2000). However, concerns have been raised (BY WHOM?) about clinical management (BY WHOM?) of advance decisions in the particular context of suicidal behaviour (MEANING?) (e.g. Dresser, 2010; Frank, 2013)."	countries, including the UK, US and Australia^{3,4} and enables patients to state their preferences for the management of their mental health condition when they may temporarily lose their mental capacity. A person with a mental health condition may also make some decisions about particular treatment that they would not wish to have and may involve an advance decision to refuse particular treatments, i.e. electroconvulsive therapy. Advance care planning has been shown to have a number of benefits in relation to healthcare for mental health patients in the UK and US, such as enhancing patient autonomy and engagement, promoting adherence to treatment plans (i.e. patients taking prescribed drugs), improving continuity of care with fewer psychiatric admissions, reducing the use of social workers' time and lower levels of violent acts.^{3,4} This increasing use of advance care planning in mental health may result in an increasing use of advance decisions to refuse mental health care treatment, and concerns about clinical management of advance decisions following self-harm and/or suicide attempts have been made by healthcare professionals and legal and ethical consultants.⁶⁻⁸
Also, it is not clear what is meant by "that decision" (page 7, line 6) and "advance decisions relating to suicidal behaviour" (page 7, line 23-24).	We have edited the text to clarify what we mean: "that decision" has been changed to "a decision about refusal of treatment" and "advance decisions relating to suicidal behaviour" has been changed to "an advance decision is appropriate for medical treatment following suicidal behaviour" We included a definition for what we meant by "advance decision" in the first paragraph
The Rationale and the Aim of the review are not clear, and should be re-phrased.	We have edited both the rationale and aim to make the context and topic of research much clearer.
Please, provide a rationale for excluding "primary conditions which were not mental health related" and define "pre-existing mental health issues" (Table 2).	We excluded studies with no existing chronic or terminal physical conditions because the focus of the review was not on end of life care. We have changed Table 2 to be in the PICOS format. We have added "with no existing chronic or terminal physical conditions" to the

	text. We note that following “primary conditions that are not mental-health related we have given examples to make it clear we are not including end of life care or physical disabilities or neurodegenerative diseases (this is what we mean by primary conditions which are not health related). “Pre-existing mental health issues” has been removed from the text in the table to improve clarity. We believe that the rationale in the text for excluding this is much clearer with the edits to the text in the introduction that we have made to make the specific context of the review clear.
Please, specify “range of disciplines” (page 10, line 27) and “factual information” (page 10, line 38).	We have added specifications and examples to the text for “range of disciplines” and “factual information”
Please, note that patients/public could have been involved in deciding whether the review topic could be informative from their perspective	We have added a more detailed statement about our PPI work relating to the paper on page 12: “An expert by-experience was a co-applicant on the NIHR Programme Grant and actively contributed to the study design and objectives. Patient advisors, carers, and clinicians evaluated the relevance and importance of the research questions for the advance decisions component of the Grant and the systematic review. Our interim and final results were presented and evaluated by clinicians, academics, patients, and carers. There was also patient input into our dissemination plan, which includes dissemination to clinicians and the relevant patient community.”
Please, provide a rationale for excluding hypothetical scenarios from some parts of the review (textual analysis?) and including such papers in other analyses (thematic analysis).	We originally had a comment relating to the rationale for the exclusion of hypothetical scenarios only in the results section, but now have it in both the method and results section and believe this makes the purposes behind removal of the hypothetical scenarios in one stage and inclusion in the other much clearer. Text has been added on page 10-11 to the method section.
The perspectives included in the thematic analysis (such as general medical and	We have edited the results of the thematic analysis so the perspectives are more clearly

psychiatry) have not been specified in a systematic manner.	delineated, clearer, and more specific and have added the following text on page 11: “In order to explore similarities and differences between disciplines, we distinguished between “general medical” as papers written from a general medical practice or emergency services perspective; “psychiatry” as those written by clinical psychiatrists or from a psychiatry perspective, “Nursing” as those written by practising nurses or research nurses, “Bioethics” as those in ethics sections in journals or written by researchers in medical ethics, “Ethics” as those in ethics journals or written by ethics researchers, and “Legal” as those written from a legal perspective and/or by a legal representative.”
It is not clear which themes identified in the review apply to all cases of clinical management of patients with advance care planning and which apply specifically to patients engaging in suicidal behaviour (e.g., promoting patient autonomy vs providing appropriate care).	We only included treatment of cases following suicidal behaviour where an advance decision. This is noted in our rationale and aims and also in the inclusion and exclusion criteria.
It is also not clear how the very complex “capacity assessment” issue should be applied to the moment an individual makes an advance care plan vs the moment of engaging in suicidal behaviour vs expectations of care after a suicide attempt. Given the current limitations of the manuscript regarding its aims and data analysis and presentation of results, I am not able to comment on Discussion of results, recommendations, future directions and conclusion.	Issues with capacity assessment were discussed both in relation to the moment a person writes an advance care plan and expectations of care after a suicide attempt. We have added the following text, “The capacity assessment was discussed in relation to when the patient is involved in advance care planning and making the decision to write an advance decision to refuse treatment. But was also discussed in relation to the judgement of capacity made by clinicians in an emergency situation, when if the person is considered to have capacity the advance decision can be ignored and they can verbally refuse/accept treatment.” On page 18 to clarify that both of these contexts were discussed and the capacity assessment was considered insufficient without an assessment of authenticity of the decision made.
Reviewer: 2	
This systematic review examining the management of patients with an advance decision and suicidal behaviour deals with an important, and ethically challenging topic. The	

review was well conducted and the manuscript is well written. I only have a few minor comments detailed below. I note that the search was conducted almost a year ago and suggest that it is updated before final submission.	We have updated the search which generated one potential source which did not meet our inclusion criteria because it was a conference abstract. The dates of searches in the text have been changed to July 2018.
In the Introduction the authors discuss the variation in legislation across nationalities and cultures. The search was not restricted to English language articles but all the studies retrieved were conducted in the UK, the USA or Australia. I suspect there would be differing views on this topic and its management in different countries, in particular countries with different implementation around this legislation, and I think this is worth coming back to briefly in the discussion.	In the original manuscript, we briefly mentioned this in the strengths and limitations section. We have retained this and added the following text in italics to make the point stronger, the text in the strengths and difficulties section now reads: Second, the articles were focussed on the US, UK and Australia, so may have resulted in bias relating to the specific legislation/ethics of those countries. There may be different views on this topic and its management in countries with different implementation of legislation, so it will be important for future research to compare findings internationally across a wider range of countries.
It might be helpful to the reader to add references to the Results section so that it is easy to see which studies the authors are referring to in each point being made. I realise this information is in the tables but as a reader it may be easier if this information was also in the text. It would also give a quick reference to how many of the studies supported each point.	We have added references from the selected studies to both the results section and the discussion and have updated the tables so we use Vancouver referencing numbering system to identify the selected studies so it corresponds with the text.
There are a couple of points made in the Results that it would be good to dig into a bit deeper in the Discussion if possible. An example being where there are similarities and differences in end of life decisions between suicide and other physical health conditions. I think this is of real interest	We have added a paragraph that discusses the differences and similarities between end of life decisions and our findings in the current review on page 21 in the discussion section.

and the challenges of balancing individual autonomy with the professional views regarding the fluctuating nature of suicidality and the treatability of mental ill-health is worth additional discussion.	We have added some additional text regarding the fluctuating nature of suicidal behaviour on page 22 in the discussion.
I also think the statement about the ‘societal expectation that suicide should be prevented’ is worth exploration in the Discussion if possible. The preventability of suicide is something that we are at pains to emphasise as a sector for a range of reasons, but perhaps there is another side to that that is worthy of some consideration.	We have added a paragraph relating to discussion about societal expectation for prevention of suicide on page 23 in the discussion.
Also in the Results section, I am not sure the term ‘technological advances’ (p21) is the right term – perhaps ‘growing evidence base’ or ‘scientific advances’ might be better.	Thankyou for pointing this out. We have changed the text from “technological advances” to “changes in evidence-base for a particular treatment or scientific advances” (now on page 19)
In the next paragraph where you state that sometimes advance decisions were reported to be ‘too specific or too general’ - I think an example or two may help the reader here i.e. what is too specific or too general when it comes to a document like this?	We have added examples of what we mean by “specific” and “general” in relation to the advance decision in brackets on page 19
At the beginning of the Discussion it may be helpful to remind the reader of how many studies were retrieved etc. I also think referencing the studies as you discuss them would be helpful throughout this section as well as in the Results.	We have now added a couple of paragraphs at the beginning of the discussion to remind the reader of the findings. We now also reference the selected studies in the discussion.
It is noted that there was evidence of differences in attitudes of clinicians from different backgrounds and I think it would be worth ensuring that this comes across clearly in the Results section.	We have added some discussion of this in the discussion section (page 21) and made the section in the results clearer (page 14).
Perhaps also provide some additional explanation as to what is meant in the Allen reference that in most cases capacity will be diminished thus adding to the complexity of advance decisions in this population. This to me seems to be very important and some additional discussion/ debate would be of interest here.	We have added the following text to that section to make this clearer: “Suicidal ideation has been associated with disordered and impulsive decision making^{33,34} and evidence indicates that most mental health patients presenting to emergency departments

	are judged as not having capacity to make a treatment decision. ¹¹ "
It would be interesting to get a sense of how the findings reported here compare to those in studies or reviews of this type of practice/legislation in the physical health sector, and if differences exist what are the implications of this for mental health patients and staff?	This is a very interesting point. Although a full discussion of advance care planning in physical health is beyond the scope of this review, we have added a paragraph that discusses the differences and similarities between end of life decisions in the context of physical illness and our findings on page 22 in the discussion section.
Overall I think this is a well-conducted study and a well-written manuscript. My the comments above really reflect that this is an important and ethically challenging topic that just warrants a bit more in-depth discussion if this is possible. As the authors note this is likely to be an increasing issue for clinicians to have to deal with, so the more detail and/or advice (including in terms of recommendations for practice and policy) that can be provided here the better. I hope these comments are helpful. Best wishes.	
Reviewer: 3	
I thought this was a very interesting article in an important area. I did however find that the introduction lacked a bit of depth and it was only when I read the results that the importance of the topic became even clearer. It would be good to include a section in the introduction why this topic is important and why we need to think about it.	We have added some text to the introduction to give it more depth, highlight background to the review and the importance of conducting the review. We have strengthened the rationale by making the context of the review clearer and by ending with the following statement to make the importance of the review clear: “A synthesis of this literature is important to examine similarities and differences and to establish the key findings, particularly as the management of advance decisions to refuse treatment of injuries and illnesses following self-harm or suicide attempts is challenging for clinicians⁸ and there is a lack of consistency of practice. A review of the literature will be important to inform guidelines for the management of advance decisions following self-harm or suicidal behaviour.”

I was also unclear about the definition of advance directives and when they were used. it would be good to provide some clarity around this. In particular in what context these would be considered (i.e after an attempt? or any other situations).	The definition we use is in the first paragraph, we appreciate that this may not be as clear as we would like it so we have strengthened the text throughout the introduction to be much more explicit about the context relating to management of an advance decision to refuse treatment following injury or illness resulting from self harm or suicide attempt.
It would also be good to include the various references in the actual result section.	We now include references to the selected studies in the text in the results section and discussion section
I also wonder if families should be included in the development of guidelines.	While that we agreed that families should be involved in writing guidelines for dealing with advance decisions in relation to suicidal attempts and self-harming behaviour, we have not mentioned the development of guidelines in the document. However, we do mention consulting with a healthcare professional when writing an advance decision so we have added consulting with families to this sentence too on page 24: "Findings indicate that it may be helpful, in this particular context, for an advance decision to be written in consultation with a professional healthcare worker and the patient's family."

VERSION 2 – REVIEW

REVIEWER	Karolina Kryszynska University of New South Wales, Australia
REVIEW RETURNED	23-Sep-2018
GENERAL COMMENTS	Thank you for submitting the revised manuscript. My concerns and suggestions have been addressed. It is an interesting and insightful review on the topic of ACP.
REVIEWER	Jo Robinson Orygen, Australia
REVIEW RETURNED	15-Sep-2018
GENERAL COMMENTS	Many thanks for addressing the comments of the reviewers so thoroughly. I agree that the paper is now stronger as a result. My only remaining suggestion is that you consider moving the definition of suicidal behaviour to earlier in the manuscript, rather

	than having it under the heading 'Aims'. You might also want to make sure that 1) you are consistent in your language about suicidal behaviour prior to presenting the definition - at the moment you still switch a bit between suicide attempt and suicidal behaviour which can be confusing. 2) you may want to acknowledge that by adopting the inclusive definition of suicidal behaviour and allowing that to encompass all types of self-harm that in fact there may be cases of NSSI that are by definition not suicidal included here. I understand the complexity of selecting a definition of suicidal behaviour so think that it is probably best to be as clear as possible as to what behavioural phenomena you are including. Just a couple of really minor points: Please spell out ED at least at first use, rather than abbreviate it (p7); Page 7 - maybe say 'bipolar disorder' rather than just 'bipolar'; and there are a couple of inconsistencies in capitalisation (p12). Many thanks
--	---

VERSION 2 – AUTHOR RESPONSE

We thank the reviewers for their useful comments on the manuscripts. Reviewer 2 did not make any suggestions of changes to the document. In response to Reviewer 1 we have moved the definition of “suicidal behaviour” to earlier in the introduction (it is now on page 7) and we have changed all references to self-harm and suicide attempts in the introduction to that terminology to be consistent. As suggested we have added a comment about non-suicidal injury to be clear to the reader that these instances are also included in the review. We have dealt with the minor grammatical issues and spelt out ED on the first instance of using it on page 7 changed “bipolar” to “bipolar disorder” on page 7 and have addressed the inconsistencies in capitalisation on page 12.